# Understanding Memorization using Representation Similarity Analysis and Model Stitching

**Aishwarya Gupta, Indranil Saha, Piyush Rai**
Department of Computer Science and Engineering
IIT Kanpur, India
{aishwaryag,isaha,piyush}@cse.iitk.ac.in

**Editors:** Marco Fumero, Clementine Domine, Zorah Lähner, Donato Crisostomi, Luca Moschella, Kimberly Stachenfeld

## Abstract

It is well-known that deep neural networks can memorize even randomly labeled training data, raising questions on our understanding of their generalization behavior. However, despite several priors efforts, the mechanism and dynamics of how and where in the network memorization takes place is still not well understood, with contradictory findings in the literature. In this work, we use representation similarity analysis methods, in particular Centered Kernel Alignment (CKA) and model stitching, as a simple but effective way to guide the design of experiments that help shed light on the learning dynamics of different layers in a deep neural networks that are trained using random (i.e., noisy) labels. Our results corroborate some of the previous findings in the literature, provide new insights into the representations learned by the layers of the network when trained with various degrees of label noise, and also provide guidance into how techniques such as model stitching could be best leveraged to understand the functional similarity of a model that has memorized with another model that has good generalization.

## 1 Introduction

Deep Neural Nets (DNN) are widely used across many domains because of their strong generalization ability on unseen inputs. However, they are also prone to memorization, i.e., they can even fit randomly labeled training datasets perfectly [25]. Memorized inputs cannot be explained using generalizable features [24], and usually represent atypical inputs in long-tailed distributions [9, 8], or the underlying noise/outliers in the training dataset [25]. Though sometimes memorization can be benign [1] and may help in generalization on downstream tasks [20], it poses privacy and security concerns [17, 4] and needs to be studied and analyzed carefully. However, the precise mechanism and dynamics of memorization are not well-understood and prior works have even reported contradictory findings, e.g., works that report that memorization happens only in deeper layers [19, 18, 2], to also works that report that memorization is dispersed and localized across many layers of the network [21].

Representation similarity analysis [14], in particular, Centered Kernel Alignment (CKA) [15], has been used to analyze and compare representations learned by different layers of the same network or of different networks. These methods have also been used in comparing generalized and memorized networks [7], and for comparing layer-wise representations of the model in problem settings such as transfer learning [10]. However, more thorough investigations of memorization in a DNN, e.g., the impact of varying levels of label randomization (i.e., degree of noise) over the learning dynamics of different layers is still lacking. In this work, we leverage CKA to compare layer-wise representations of a DNN with high generalization and negligible memorization to another DNN with weaker

generalization and high memorization. We simulate this by training a model on the clean dataset (referred to as a clean/noise-free model) and a dataset comprising a fraction of randomly labeled inputs (referred to as a noisy model) respectively. This helps us gain insights into the (dis)similar learning dynamics between different layers of generalized and memorized models. Moreover, we further investigate the functional similarity [3, 13] of these layers by stitching early layers of the generalized model to the latter layers of the memorized model and vice versa [6]. This provides insights into the representational and functional aspects of these layers that can enhance the understanding of memorization and can help formulate methodologies to mitigate and fix it.

## 2 Background

Some of the previous works on understanding memorization in DNNs have used methods such as prediction depth [23] of each input, and manifold analysis techniques [5]. In contrast, we leverage representation similarity measures which provide a simple way to compare different layers of the same network or of two different networks. These measures provide a simple method to track how the layers evolve and eventually converge. In particular, we use Centered Kernel Alignment (CKA) [15] which is computed using the normalized Hilbert Schmidt Independence Criterion (HSIC) [11] for comparing representations of two layers from the same network or from two different networks. For any two layers $l$ and $k$ of sizes $d_l$ and $d_k$, respectively, with their corresponding layer activations denoted by $X \in \mathbb{R}^{N \times d_l}$ and $Y \in \mathbb{R}^{N \times d_k}$ on a dataset $\mathcal{D}$ of size $N$, CKA is defined as $\text{CKA}(l,k) = \frac{\text{HSIC}(X,Y)}{\sqrt{\text{HSIC}(X,X),\text{HSIC}(Y,Y)}}$, where $\text{HSIC}(X,Y) = \frac{1}{(N-1)^2}\text{trace}(XHYH)$, and $H = I_N - \frac{1}{N}\mathbf{1}\mathbf{1}^T$ denotes the centering matrix used to center the matrices $X$ and $Y$.

In addition to CKA, we also leverage model stitching [3] which can also be used to analyze representations learned by DNNs. Stitching connects the bottom layers of one network with the top layers of another network with a simple trainable layer in between, with the performance boost/drop indicating the degree to which the representation learned by the two models are similar to each other.

## 3 Experiments

To understand the dynamics of memorization in a DNN, we experiment with two different architectures - a customized CNN and ResNet-18 [12] model trained on the SVHN [22] and CIFAR-10 [16] datasets, respectively. We create noisy versions of each dataset by randomly sampling $r = \{5\%, 10\%, 15\%, 20\%, 25\%, 37.5\%, 50\%, 67.5\%\}$ percentage of total images from its training dataset and assigning them random labels. We then train a model for each value of $r$, including $r = 0$ (i.e., clean train dataset) as a noise-free baseline model. During each noisy model's training, checkpoints are saved at regular intervals and are used to study layer-wise memorization. In all our experiments, we use the test split of the dataset to compute the CKA metric. Further training details are mentioned in the supplementary material.

### 3.1 Convergence of different layers trained with varying fraction of random labels

In this experiment, we study the convergence of different layers of the network as the noise ratio $r$ increases in the training data by evaluating the similarity of the intermediate checkpoints with the final trained model. For each value of $r$, the similarity between the $i^{th}$ layer of the saved checkpoints and the corresponding $i^{th}$ layer in the final trained model is computed using CKA. In Figure 1, we show the convergence of ResNet-18 model trained on CIFAR-10 dataset for varied $r$. Note again that $r = 0$ represents the model trained on the clean dataset. The convergence of CNN models trained on noisy SVHN dataset is included in the supplementary.

As shown in Figure 1, the early layers converge quickly whereas the deeper layers converge slowly. This is consistent with what is observed in earlier works as well [5, 23]. Moreover, *irrespective of the noise ratio $r$, the convergence of the early layers follows a similar trend to that of the noise-free model, suggesting a low influence of data noise on these layers*. In contrast, the convergence rate of the deeper layers is directly proportional to the noise ratio; the models trained with large $r$ exhibit the slowest convergence. Based on these observations, *we hypothesize that irrespective of $r$, the early layers of the noisy models are mostly functionally similar to the early layers of the noise-free model, suggesting low memorization for the very early layers*.

To validate this hypothesis, we leverage model stitching [6] to evaluate functional similarity. For each value of $r$, we create a stitched model by stitching the early $k$ layers of the noisy model to the latter layers of the noise-free model starting from the $(k + 1)^{th}$ layer using either a $1 \times 1$ convolution or

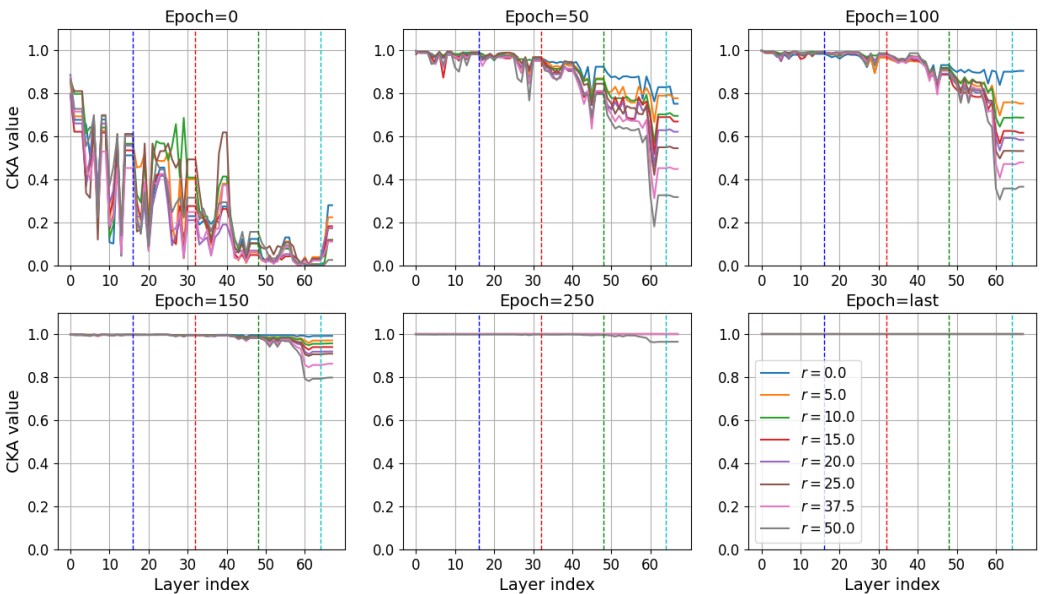

Figure 1: Layer convergence of ResNet-18 model trained on CIFAR-10 dataset for varied noise ratio $r$. The four vertical dashed lines mark the last layer of each residual block.

linear layer. Only this additional layer is trained to maximize the accuracy of the stitched model on the clean dataset. If the layers of the noisy model are functionally similar to that of the noise-free model, then the performance (accuracy) of the stitched model should not drop much, suggesting high functional similarity and low influence of label noise on the layers of the noisy model contained in the stitched model. We experiment on CIFAR-10 dataset by stitching the first $k \in \{1, 2, 3, 4\}$ residual blocks of the noisy ResNet-18 model with the latter blocks of the noise-free model, for all noise levels $r$. Similarly, for SVHN, we stitch up to every convolution and linear layer of the model for all values of $r$. We present our findings in the first column of Figure 2. The X-axis represents noise percentages $r$ in the training dataset and the Y-axis shows the test accuracy of the stitched model. Each line plot corresponds to a stitched model formed by stitching layers up to the respective residual block/layer of the noisy model to the noise-free model. As observed, irrespective of the $r$ value, stitching early blocks/layers of all the noisy models results in only a marginal drop in the accuracy of the noise-free model. However, the accuracy starts dropping substantially with stitching more layers of the noisy model and the performance drop is proportional to the noise $r$. This result affirms that the early layers of the noisy model have high functional similarity to the corresponding layers of the noise-free model, suggesting low memorization of random noise present in the training dataset. However, as the depth of the noisy model increases in the stitched model, the performance starts dropping suggesting low functional similarity with the clean model's layers and a high influence of noise.

### 3.2 Fix where? Early layers or deeper layers or maybe a few others too?

Next, we perform another layer-wise comparison between the noisy and noise-free models to further analyze memorization and potential ways to mitigate it. For all values of $r$, we compute the CKA score between corresponding layers of the noisy model at varied checkpoints and layers of the finally trained noise-free model and report the findings in Figure 3. The figure shows that, during the early training phase (e.g., epoch 50), all layers of all the models learn representations that are similar to those learned by the noise-free model. However, as training progresses, the representation similarity between corresponding layers of the noisy and noise-free models starts decreasing and the difference becomes prominent in deeper layers. The drop in similarity is proportional to the noise ratio present in their respective training datasets. This is consistent with the previous findings of early stopping [18] and the efficacy of many early learning regularization approaches [19, 2] in mitigating the impact of noise.

To investigate further, we apply the stitching idea again but, this time, stitch the early layers of the noise-free (clean) model with the deeper layers of the noisy model. As shown in the second column of Figure 2, stitching early layers/blocks of the clean, noise-free model to latter layers/blocks of the

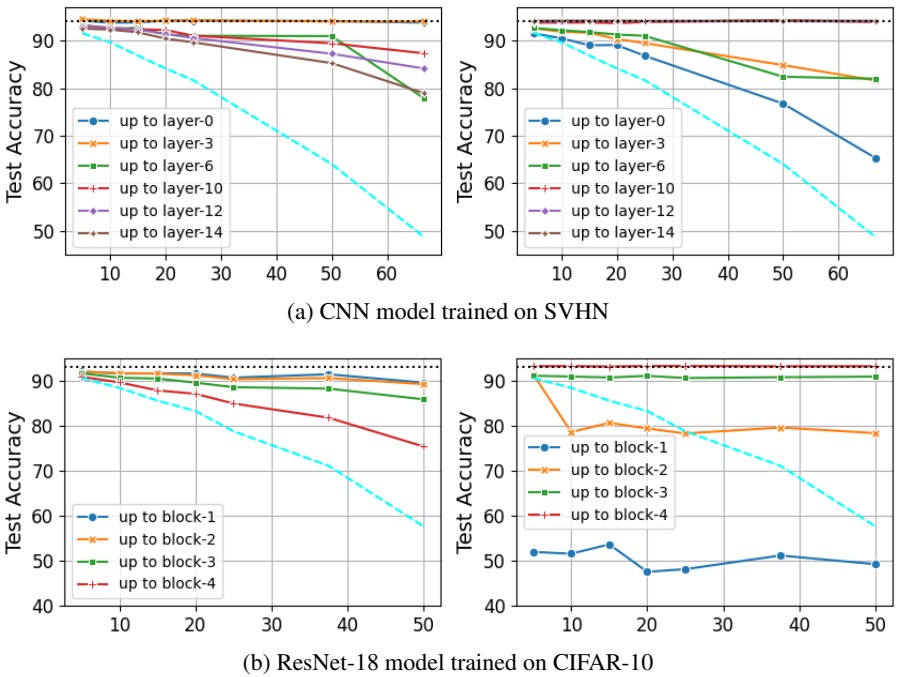

(a) CNN model trained on SVHN

(b) ResNet-18 model trained on CIFAR-10

Figure 2: Test performance of the stitched models. The left plot corresponds to stitching the first $k$ blocks/layers of the noisy model to a noise-free model. Similarly, the right plot shows the stitching of the first $k$ blocks/layers of a noise-free model to a noisy model. The black dotted line and the cyan dashed line show the test performance of the clean and noisy models respectively.

noisy models does not improve the stitched model's performance, corroborating the fact that early layers of noisy models do not have significant memorization. However, stitching the clean model up to half its depth to the noisy model shows performance improvement. For instance, for ResNet-18 model, stitching the noise-free model up to the second residual block improves the noisy model's performance substantially; with further stitching resulting in performance at par with that of the noise-free model, irrespective of the $r$ value. The similar trend is observed for CNN model trained on noisy SVHN dataset. (We further validate our finding by computing the similarity between the layers of the stitched and clean model and discuss it in the supplementary material). However, the converse does not hold i.e., stitching a noisy model up to half of its depth to the clean model does not degrade its performance substantially as shown by the "blue" and "orange" colored lineplots in the left subplot of Figure 2(b). This shows that, unlike conventional wisdom, fixing only the last layers may be insufficient to mitigate memorization. Instead, fixing a few of the early layers of the noisy model could also contribute in helping mitigate memorization, potentially because of a cascaded overall effect these early layers might also have (as opposed to the relatively less significant effect of fixing just the end layers), and help achieve high generalization performance similar to the clean model. This experiment indicates that the effect of memorization may actually also be present in the early layers and resonates with the recent insights that memorization may not be limited to the last layers but is localized in a subset of neurons across multiple layers (including the early layers) of the model [21]. Thus the difference in the representations learned in the last layers of a memorized network as compared to that of a generalizable network could possibly be due to a cascading of subtle but important differences in earlier layers' representations too, and not solely due to the last layers learning a different representation all by themselves.

## 4 Conclusion

Using representation similarity analysis (in particular, CKA) and model stitching, we have analyzed the learning dynamics of various layers of DNNs that memorize because of being trained on randomly labeled data. Our empirical analysis both corroborates earlier findings in the literature, as well as provides new insights. In particular, it sheds light on how the early and the deeper layers of the network evolve during training, the effect of the amount of noise in the labels on their convergence, and also provided insights about which parts of the networks should be fixed to mitigate memorization.

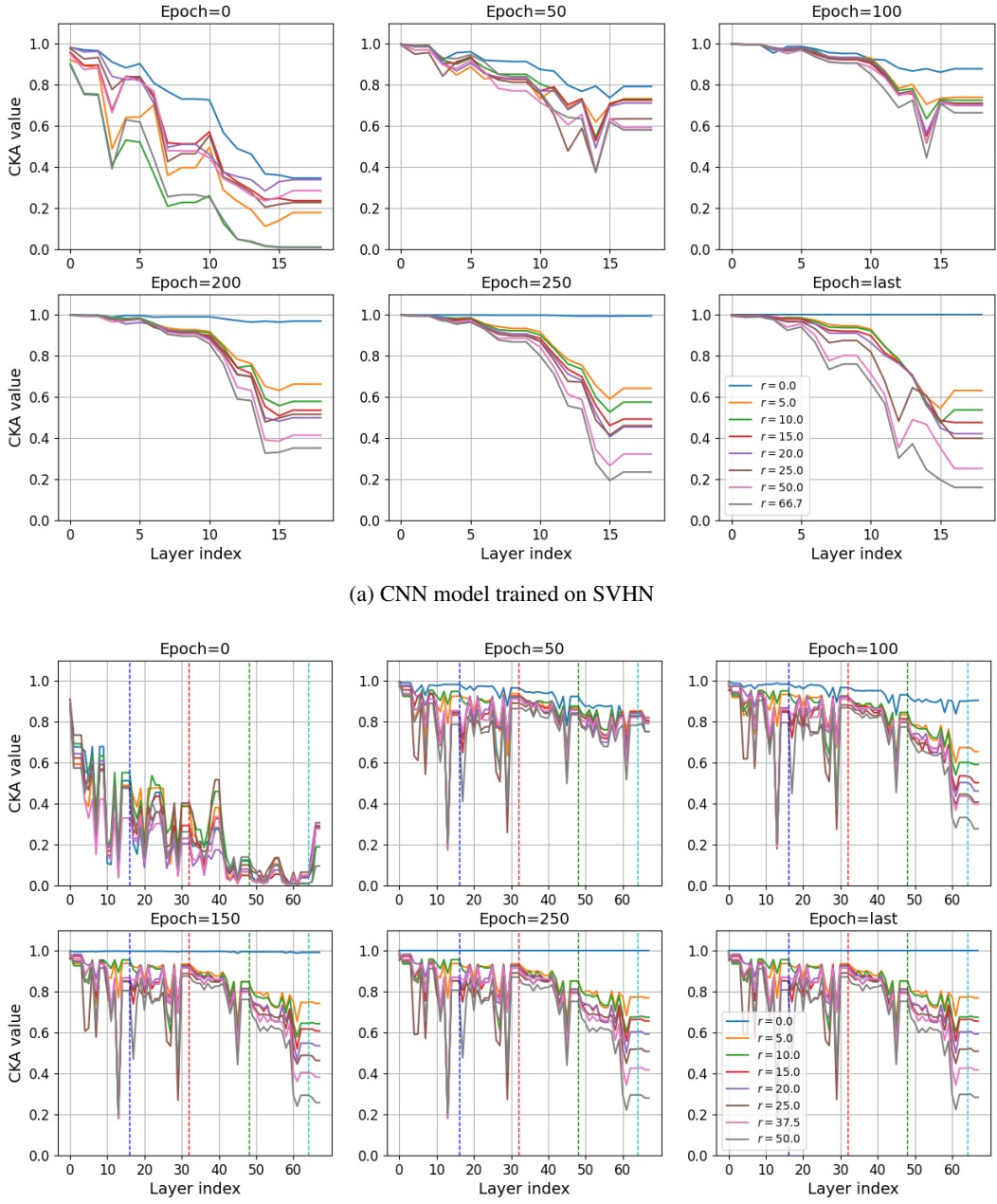

(a) CNN model trained on SVHN

(b) ResNet-18 model trained on CIFAR-10. The four vertical lines correspond to each residual block.

Figure 3: Comparison of noisy models' layers to the layers of the final trained noise-free model.

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
