# Supplementary: Understanding Memorization using Representation Similarity Analysis and Model Stitching

**Editors:** Marco Fumero, Clementine Domine, Zorah Lähner, Donato Crisostomi, Luca Moschella, Kimberly Stachenfeld

## 1 Experiments

### 1.1 Setup

We experiment on two datasets - SVHN and CIFAR-10 using two different DNN architectures. We train a customized CNN model on the SVHN dataset and a ResNet-18 model on the CIFAR-10 dataset. The CNN model comprises three convolution layers and three linear layers with maxpool layer after each convolution layer and uses ReLU as an activation function. For each dataset, we experiment with multiple noise percentage $r$, specifically setting it to a wide range of values such as $\{5\%, 10\%, 15\%, 20\%, 25\%, 37.5\%, 50\%, 62.5\%\}$. For all values of $r$, the model is trained using an SGD optimizer with an initial learning rate of $0.1$ and decaying it by a factor of $0.1$ whenever training loss plateaus. The training is stopped only if the training accuracy is at least $98\%$; ensuring memorization of noisy training inputs. The training (both on clean and noisy training inputs) and test performance of all the models including the baseline noise-free ($r = 0$) model with training epochs is shown in Figure 1.

#### 1.1.1 Convergence of Different Layers Trained on Noisy Data

For each model, the similarity between layers at any intermediate checkpoint and layers of the final trained model is measured using CKA. For all values of $r$, the CKA similarity value is computed between the layers of the model saved after $e \in \{0, 50, 100, ..., 700\}$ training epochs and the layers in the finally trained model. A high similarity value shows the convergence of the layer. The result of the CNN model trained on SVHN dataset is shown in Figure 2.

#### 1.1.2 Similarity between the Layers' of Noisy Model and Noise-free Model

In this experiment, we compare the representational similarity of layers of noisy models to the layers of noise-free models. For all noise percentages $r$, the CKA similarity value is computed between the layers of the noisy and noise-free model to understand the similarity of a noisy model (high memorization) to a model with high generalization. We evaluate the CNN model trained on the SVHN training dataset with varied noise percentage $r$ at multiple checkpoints and report the results in Figure 3.

Next, to support the findings of the stitching experiment, we further compare the CKA similarity of the layers of the noisy model in the stitched model to the layers of the noise-free model. The stitched model is obtained by stitching up to $k$ residual blocks of noise-free model to latter blocks of noisy model trained on CIFAR-10 dataset for varied noise percentage $r$. A layer with high similarity is expected to have similar representations to that of the corresponding layer in the generalizable model.

Proceedings of the II edition of the Workshop on Unifying Representations in Neural Models (UniReps 2024).

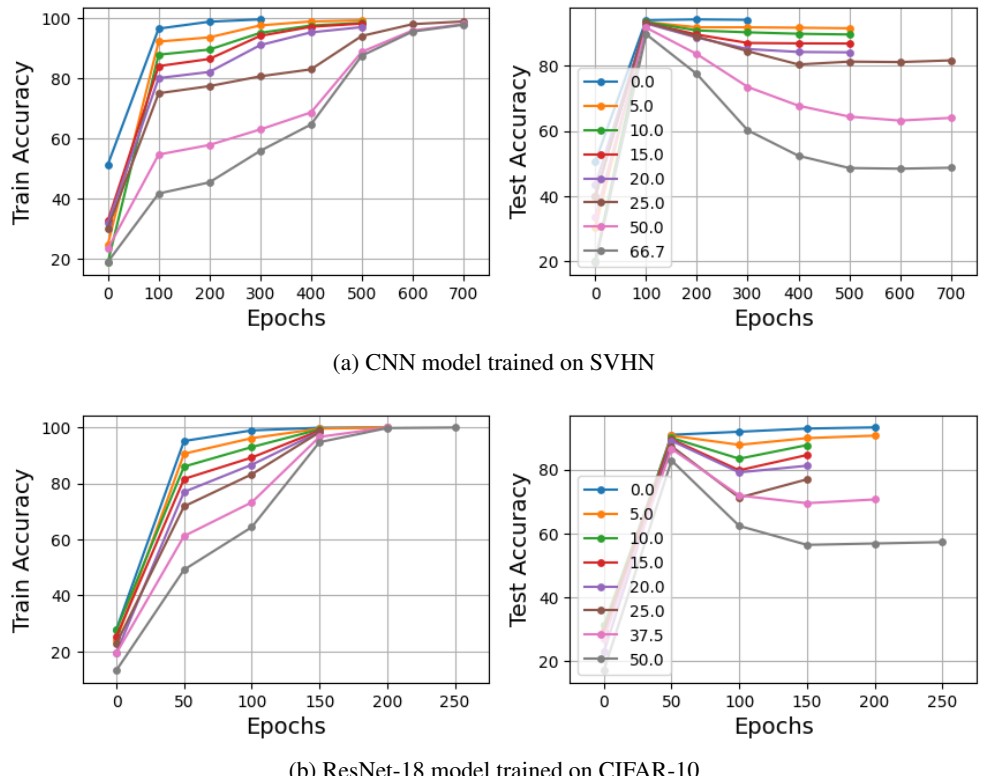

(a) CNN model trained on SVHN

(b) ResNet-18 model trained on CIFAR-10

Figure 1: Train and test performance of noisy and noise-free models evaluated at multiple training epochs.

The CKA similarity values are shown in Figure 4 for different stitched models obtained by stitching layers up to $k$ blocks of the noise-free model to the latter layers of the noisy model.

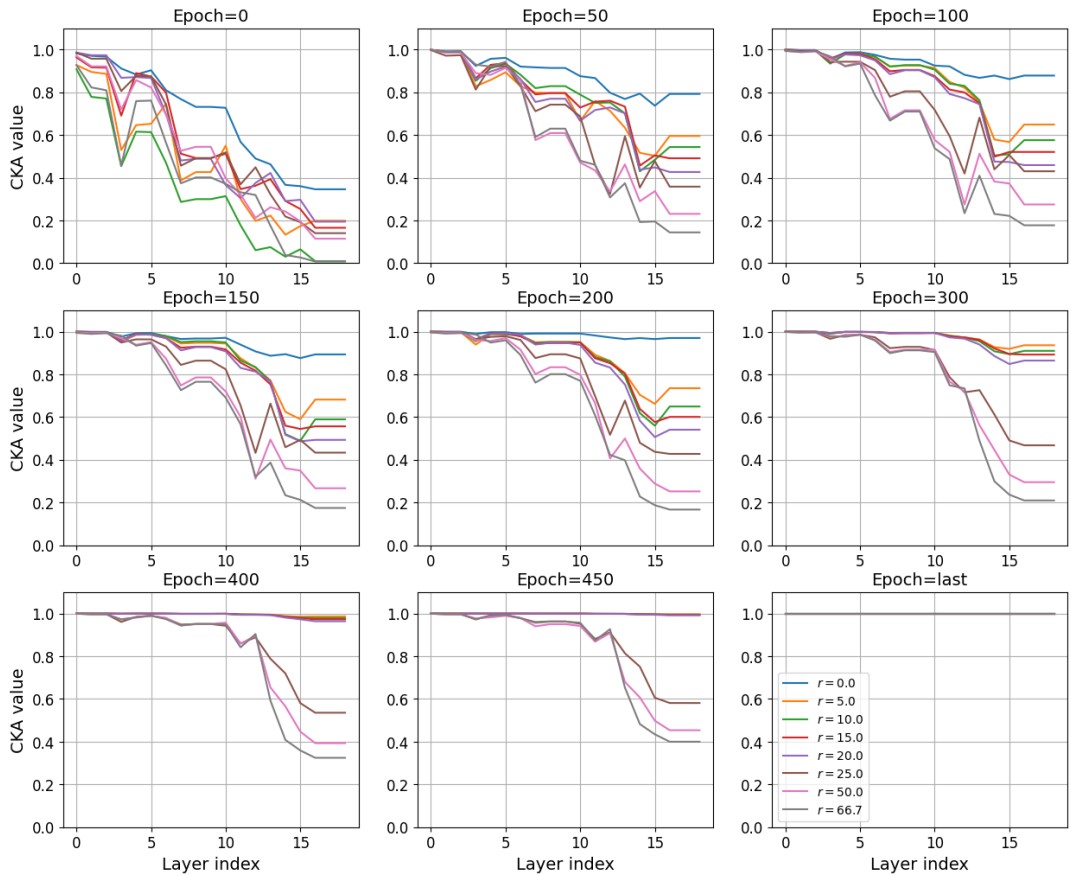

Figure 2: Convergence of layers of CNN model trained on SVHN dataset for varied noise ratio in the training dataset.

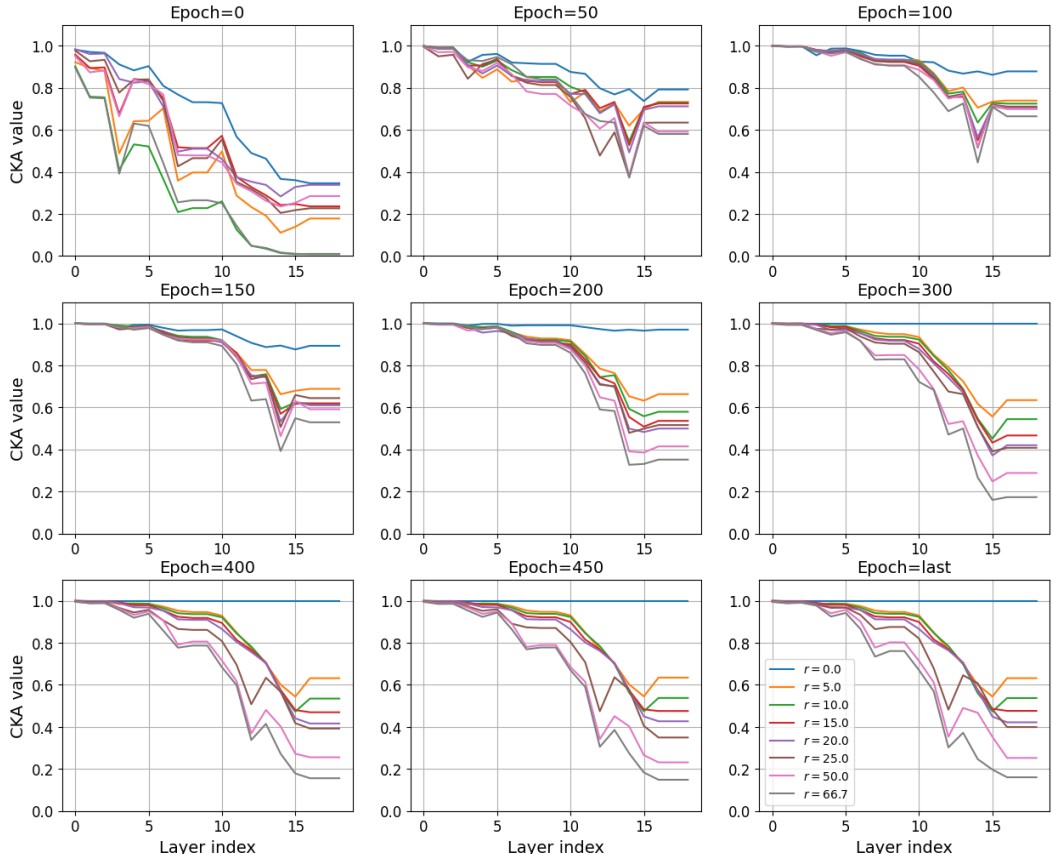

Figure 3: CKA-based comparison of the representation similarity between the layers of the noisy model and the layers of the final trained noise-free model. Each subplot corresponds to a specific checkpoint during the training and shows a similarity of layers of the noisy model at the given checkpoint to the layers of the final trained noise-free model.

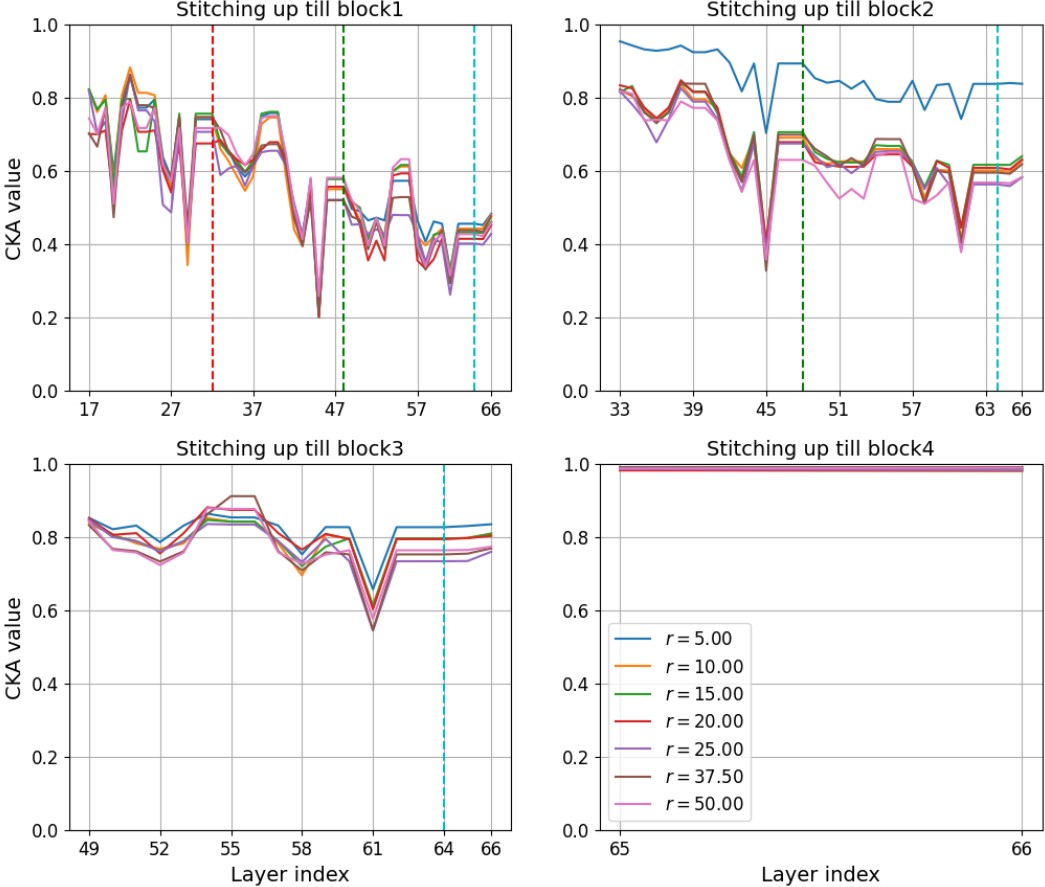

Figure 4: Similarity of the layers of the noisy model in the stitched model to the noise-free model. Stitching is done by stitching early layers of the noise-free model to late layers of the noisy model trained on CIFAR-10 dataset for varied noise percentage $r$.