# OpenReview forum: "Understanding Memorization using Representation Similarity Analysis and Model Stitching"
_NeurIPS.cc/2024/Workshop/UniReps — UniReps_

### Official Review · Reviewer_Y7Uk · 2024-09-27

**Rating:** 7
**Confidence:** 4

**Review:**

This paper investigates the mechanisms of memorization in deep neural networks (DNNs) trained with random labels. The experiments use CKA and to compares layer-wise representations between DNNs with high generalization and low memorization against those with weaker generalization and high memorization. The results show that early layers converge quickly and are less influenced by label noise, while deeper layers exhibit slower convergence and greater sensitivity to noise levels. Interestingly, the research demonstrates that fixing early residual blocks can effectively mitigate memorization without significantly compromising model accuracy, challenging the traditional focus on only the final layers.

Strengths:
- Interesting findings. Shows that early layers are less impacted by memorization than deeper layers from a representation similarity perspective. Also shows how representation similarity changes over the course of training.
- Challenges the conventional wisdom re: fixing only the last layers suffices to mitigate memorization.
- Use of model stitching to supplement CKA findings. I like the use of model stitching for validating trends/findings identified via the CKA representation similarity analysis, which by itself can be unreliable [4].

Weaknesses:
- Novelty. As mentioned in the paper, there are quite a few papers that study memorization from a representation similarity perspective. So, a discussion contrasting the findings in this paper more clearly to specific findings in previous work would be useful.

- Analysis of memorization specific to random labels setting. To me, it is unclear whether these findings are specific to memorization induced via random labels or whether they are more general and can be observed in other more realistic settings as well [2,3].

Suggestions for improvement:
- Would be interesting to compare models trained with different levels of random label noise at a more fine-grained example level. One way to do this is to use similarity metrics based on data attribution (e.g., ModelDiff [5] or Influence embeddings [6]) to study how model predictions of generalizable DNN and memorization DNN functionally differ at the example level. Does memorization completely change how a model relies on individual training examples?


[1] MohammadReza Davari, Stefan Horoi, Amine Natik, Guillaume Lajoie, Guy Wolf, and Eugene Belilovsky. Reliability of cka as a similarity measure in deep learning. In The Eleventh International Conference on Learning Representations

[2] Carlini, N., Ippolito, D., Jagielski, M., Lee, K., Tramer, F. and Zhang, C., 2022. Quantifying memorization across neural language models. arXiv preprint arXiv:2202.07646.

[3] Somepalli, G., Singla, V., Goldblum, M., Geiping, J. and Goldstein, T., 2023. Understanding and mitigating copying in diffusion models. Advances in Neural Information Processing Systems, 36, pp.47783-47803.

[4] Davari, M., Horoi, S., Natik, A., Lajoie, G., Wolf, G. and Belilovsky, E., 2022. Reliability of cka as a similarity measure in deep learning. arXiv preprint arXiv:2210.16156.

[5] Shah, H., Park, S.M., Ilyas, A. and Madry, A., 2023, July. Modeldiff: A framework for comparing learning algorithms. In International Conference on Machine Learning (pp. 30646-30688). PMLR.

[6] Wang, F., Adebayo, J., Tan, S., Garcia-Olano, D. and Kokhlikyan, N., 2024. Error discovery by clustering influence embeddings. Advances in Neural Information Processing Systems, 36.

---

### Official Review · Reviewer_ugNL · 2024-10-05
**A Comprehensive Analysis of model memorization**

**Rating:** 6
**Confidence:** 4

**Review:**

This paper provides a novel investigation into the dynamics of memorization in deep neural networks (DNNs), specifically using two core techniques: Centered Kernel Alignment (CKA) and model stitching. The paper successfully addresses gaps in the literature about whether memorization is confined to certain layers or dispersed throughout the network, offering new insights into how DNNs behave under noisy training conditions.

**Strengths:**
1. **Innovative Use of CKA and Model Stitching:**

The paper leverages CKA and model stitching technique, providing experimental support for conclusions about functional similarities between noisy and noise-free networks.

2. **Thorough Experiments:**

The experiments use varied noise levels on the CIFAR-10 dataset, and the findings are backed by multiple checkpoints and extensive layer-wise analysis.

3. **Addressing Prior Contradictions:**

The paper successfully reconciles conflicting findings in the literature. It shows that memorization can indeed be dispersed, but is more significant in deeper layers, providing a clearer understanding of where and how memorization occurs.

**Weaknesses:**
1. **Limited Applicability Beyond CIFAR-10:**

While the findings are compelling, the scope of the dataset (CIFAR-10) is somewhat narrow. Extending the experiments to more complex datasets like ImageNet or tasks beyond image classification would provide better generalization of the results.

2. **Stitching Not Fully Explored:**

As shown in Figure 2, for a model trained under the same noisy percentage, respectively replace the first one/two layers and the last one/two layers with the corresponding layer of the noise-free trained model, and it can be found that the test accuracy rate of the latter is always higher than that of the former, which seems to be in contradiction with the conclusion in the article (L129-137).

3. **Lack of Conceptual Explanations:**

The paper lacks clear definitions for key concepts such as memorization and generalization, which are crucial for the reader to fully grasp the contributions. Additionally, CKA and other methods, while used effectively, are not explained in sufficient detail for readers unfamiliar with these techniques.

4. **Minor Typos:**

There are several minor errors in the paper, such as a typo in line 93 where it should refer to "Figure 2(a)" instead of "Figure 3(a)." Additionally, the legends in Figure 2 are inconsistently labeled with different capitalizations, which could confuse readers.

**Originality and Significance:**

The paper makes a significant contribution by improving our understanding of memorization in DNNs through representation similarity and model stitching. These tools offer a fresh perspective on comparing noisy and noise-free models, leading to actionable insights into the learning dynamics of DNN layers. Additionally, the insights into functional similarity between layers from different networks could be useful in designing more robust and generalizable architectures.

**Clarity:**

The paper is well-written and clear, with logical flow and detailed descriptions of the methods. The figures and plots effectively illustrate key points. However, certain technical aspects of CKA and model stitching may benefit from additional explanation for readers less familiar with these techniques.

---

### Official Review · Reviewer_Zpq1 · 2024-10-06
**Paper provides confirmatory understanding of older findings using CKA; model-stitching gives better insight**

**Rating:** 9
**Confidence:** 4

**Review:**

**Summary**
- This paper investigates the dynamics of memorization in deep neural networks (DNNs) using Centered Kernel Alignment (CKA) and model stitching techniques. The authors train ResNet18 on CIFAR-10 with varying degrees of label noise and analyze the convergence and similarity of layer representations. Findings reveal early layers extract generalizable features, whereas memorization appears localized to deeper layers.

**Strengths**
1. Strong clarity of method. Experimental procedure is elegant, simple and clearly explained.
2. Valid use of representation similarity (CKA) to gain understanding of where DNN memorization happens
3. The model stitching experiment provides improved insight in a causal / intervention-based way

**Weaknesses**
1. While the paper's findings regarding CKA and memorization in deeper layers are insightful, they are not entirely novel, as similar procedures have been used in previous works (e.g. ref 22). However, the confirmation of these results using different methods does add value to the field. The paper's main interesting and original contribution lies in the model stitching experiment, which challenges prevailing notions by showing that mitigations against memorization should occur in early layers too
2. It would have been nice to offer more mechanistic proposals on why the CKA method and model-stitching methods arrived at seemingly different conclusions, but since this is an extended abstract the brevity is understandable

---

### Official Review · Reviewer_uSSz · 2024-10-07
**Late layers memorize more compared to early ones**

**Rating:** 6
**Confidence:** 4

**Review:**

This paper explores the dynamics of memorization in deep neural networks (DNNs), focusing on how layers in these networks behave when trained with noisy data. Authors use Centered Kernel Alignment (CKA) and model stitching as tools to analyze representations learned by neural networks and the learning dynamics at various layers. They compare models trained on clean data and models trained on varying levels of label noise. They conducted experiments on ResNet-18 using the CIFAR-10 dataset, varying the noise in the training labels to study how different layers of the network were affected. The findings suggest that early layers are less affected by noise and retain functional similarity to clean models, while deeper layers are more prone to memorization. They also propose that fixing early layers can help mitigate memorization.

**Strengths:**

Novel Approach to Memorization: The paper brings in CKA and model stitching to analyze memorization in neural networks, which is a fresh approach.

Important Findings on Early and Deep Layers: Authors observed that early layers are less prone to memorization and remain functionally similar to clean models.

Practical Implications: The suggestion that fixing early layers can help mitigate memorization is a valuable insight for researchers and practitioners dealing with noisy datasets. It opens up new possibilities for improving model generalization in the presence of noise.


**Weaknesses:**

Limited Dataset and Architecture Choices: While the use of CIFAR-10 and ResNet-18 is standard for many experiments, these choices might limit the generalizability of the findings. Memorization behaviors might differ with larger or more complex datasets, or different architectures (e.g., transformers or more recent models). Expanding the experiments to other datasets and models would strengthen the conclusions.

---

### Decision · Program_Chairs · 2024-10-10

**Decision:**

Accept

**Comment:**

In light of the positive reviewers' feedback and relevancy of the submission, we are pleased to accept this paper for presentation at UniReps 2024. We kindly ask the authors to incorporate the reviewers' suggestions and feedback in the final camera-ready version of the manuscript.